# Prognosis and Survival in Idiopathic Pulmonary Fibrosis in the Era of Antifibrotic Therapy in Italy: Evidence from a Longitudinal Population Study Based on Healthcare Utilization Databases

**DOI:** 10.3390/ijerph192416689

**Published:** 2022-12-12

**Authors:** Marica Iommi, Andrea Faragalli, Martina Bonifazi, Federico Mei, Lara Letizia Latini, Marco Pompili, Flavia Carle, Rosaria Gesuita

**Affiliations:** 1Center of Epidemiology Biostatistics and Medical Information Technology, Department of Biomedical Sciences and Public Health, Università Politecnica delle Marche, 60121 Ancona, Italy; 2Department of Biomedical Sciences and Public Health, Università Politecnica delle Marche, 60121 Ancona, Italy; 3Respiratory Diseases Unit, Azienda Ospedaliero-Universitaria “Ospedali Riuniti”, 60166 Ancona, Italy; 4Regional Health Agency of Marche, 60121 Ancona, Italy; 5National Centre for Healthcare Research and Pharmacoepidemiology, 20126 Milano, Italy

**Keywords:** secondary health databases, acute exacerbation, survival analysis, real-world evidence, idiopathic pulmonary fibrosis

## Abstract

The aim was to evaluate the determinants of acute exacerbation (AE) and death in new cases of idiopathic pulmonary fibrosis (IPF) using administrative databases in the Marche Region. Adults at their first prescription of antifibrotics or hospitalization with a diagnosis of IPF occurring in 2014–2019 were considered as new cases. Multiple Cox regression was used to estimate the risk of AE and of all-cause mortality adjusted by demographic and clinical characteristics, stratifying patients according to antifibrotic treatment. Overall, 676 new cases of IPF were identified and 276 deaths and 248 AE events occurred. In never-treated patients, the risk of AE was higher in patients with poor health conditions at diagnosis; the risk of death was higher in males, in patients aged ≥75 and in those with poor health conditions at baseline. The increasing number of AEs increased the risk of death in treated and never-treated patients. Within the limits of an observational study based on secondary data, the combined use of healthcare administrative databases allows the accurate analysis of progression and survival of IPF from the beginning of the antifibrotic therapy era, suggesting that timely and early diagnosis is critical to prescribing the most suitable treatment to increase survival and maintain a healthy life expectancy.

## 1. Introduction

Idiopathic pulmonary fibrosis (IPF) is a rare, devastating, fibrosing lung disease of unclear etiology, although cellular senescence and microinjuries due to environmental exposure (to metal, wood dust, stone, sand and silica, working in agriculture and farming), lifestyle (current or a history of smoking), microbial agent infections (viral, fungal and bacterial) and a predisposed genetic background are currently deemed to play a relevant role in pathogenesis of IPF [1].

Although data on IPF prevalence and incidence are heterogeneous worldwide, above all depending on differences in diagnostic procedures or in the data sources used to identify IPF cases [2], a general consensus has been reached on the fact that IPF mainly affects males and older people over 60 years of age [3,4,5,6,7].

IPF is characterized by a progressive worsening of dyspnea and functional decline, with a median survival of 3–5 years from diagnosis [8,9]. However, a high variability in disease progression has been observed, with patients experiencing a rapid decline and some others progressing more slowly [9].

One of serious events that drastically worsens patients’ health conditions is acute exacerbation (AE). Its incidence varies between 4 and 24 per 100 patient-years worldwide, depending on the study design (clinical trial-based vs population-based design) or on the criteria used to define AE [10]. According to the latest expert statement on this topic, the current AE definition is “An acute, clinically significant respiratory deterioration characterized by evidence of new widespread alveolar abnormality” [10]. This serious event has been found to precede deaths and reduces the median survival time to approximately 3 or 4 months [11,12].

Moreover, patients with IPF are at higher risk of developing lung cancer compared to non-IPF subjects; in a recent meta-analysis, the estimated incidence rate ratio was 6.42 (95% CI, 3.21–9.62) and the global incidence rate in patients with IPF was 2.07 per 100 person-years (95% CI, 1.46–2.67) [13]. The recent approval—with conditional recommendation [14]—of specific antifibrotic drugs for IPF treatment (Pirfenidone and Nintedanib) has hugely changed the landscape of therapeutic management and has improved patient prognosis [9,15]. Despite the fact that the effectiveness of antifibrotic treatment in reducing lung function decline and exacerbation is well established, the current percentage of treated patients remains low with only a slight increase over time [15,16,17].

Several studies, both randomized clinical trials and observational studies, have analyzed the survival and the occurrence of AE in subjects with IPF from the time when specific drugs were introduced to treat the disease in the real world, but few were population-based [18]. Healthcare administrative databases are valuable secondary sources for estimating health outcomes as they record all clinical events or access to health care services of an unselected population. The link between different healthcare databases allows the depiction of the evolution in individuals’ health conditions [19].

The aim of the study was to evaluate the demographic and clinical factors associated with the risk of AE and of death in patients aged ≥18 years with a new diagnosis of IPF, residing in the Marche Region, during the period of the study (1 January 2011–31 December 2019), using real-world data.

## 2. Materials and Methods

### 2.1. Study Population, Data Sources, Incident IPF Case Definition

Study population, data sources and IPF case definition were previously described in detail [7]. Briefly, this is a longitudinal prospective study, based on the healthcare administrative databases, considering as the target population all adult beneficiaries of the National Health Service (NHS) residing in the Marche Region between 1 January 2011 and 31 December 2019.

Four administrative regional healthcare databases were used as secondary sources: the Regional Beneficiaries Database (RBD), Hospital Discharge Records (HDR), Drug Prescriptions (DP) Database, and the Outpatient Care Database (OCD). The beneficiary’s identification code was used to link the four databases, according to the Italian Personal Data Protection Law. 

Incident IPF cases, occurring between 1 January 2014 and 30 June 2019, were identified using the General Case Definition algorithm [7], including all individuals at their first hospitalization with a diagnosis of IPF (ICD-9-CM code 516.3), adding all individuals at their first drug prescription of Pirfenidone (ATC: L04AX05) or Nintedanib (ATC: L01XE31—L01EX09). Cases residing in the Marche Region for less than 3 years before the date of IPF identification or with hospitalization or a drug prescription for IPF between 2011–2013 (the wash-out period) were excluded. Moreover, all cases with a previous diagnosis of pulmonary malignant neoplasm occurring three years before the IPF identification were excluded. Pulmonary malignant neoplasm was detected using HDR, the DP database and the OCD (Appendix A).

IPF-patients were followed from the date of the first hospitalization with a diagnosis of IPF or the first antifibrotic drug prescription (index date) to 31 December 2019.

### 2.2. Variables

Sex, age, and comorbidities were evaluated at the index date. Age was dichotomized at 75 years; comorbidities were evaluated using the Multisource Comorbidity Score (MCS) [20], a score based on HDR and DP databases, that measures patients’ health status in the two years preceding the index date and was considered in two classes (0–4, good or fair health conditions; ≥5, poor health conditions). Patients with at least one prescription of antifibrotics during the follow-up were considered as treated.

### 2.3. Outcomes

All-cause mortality was obtained by collecting the date of death from the RBD. In addition, the in-hospital mortality, i.e., all deaths occurring up to 7 days from the date of discharge from the first hospitalization with diagnosis of IPF, was evaluated in all new cases detected by HDR.

AE was defined as any acute respiratory-related hospitalization occurring after the index date. Appendix A shows the list of ICD-9-CM codes in the HDR database’s primary or secondary diagnoses.

The pulmonary malignant neoplasm occurred after the IPF diagnosis was analyzed as an outcome related to IPF.

### 2.4. Statistical Analysis

Summary statistics were used to describe the study cohort, using mean and standard deviations or median and interquartile range (IQR) for continuous variables, according to their distribution. Absolute and percentage frequencies were used to summarize qualitative variables. Mean cumulative counts from 1 up to 5 years of follow-up were estimated to evaluate the total burden of subsequent AEs among IPF patients by treatment [21], considering death as a competing risk event; comparisons of mean cumulative count curves were performed via the area under the curve (AUC) [22].

In-hospital all-cause mortality was estimated by calculating the 95% confidence interval (95% CI) of the proportion of deaths of new IPF cases identified by HDR, using the binomial distribution.

Survival analysis was used to estimate the outcomes, using the Kaplan–Meier product limit estimator.

When evaluating AE events, death was considered as a competing risk event; patients were followed from the index date to the earliest date between AE, death, emigration or 31 December 2019; whereas, in the survival analysis, IPF cases were followed from the date of IPF incidence to the date of death, to emigration or to 31 December 2019, whichever came first.

All patients were followed up for at least 180 days. Kaplan–Meier curves were estimated, stratifying by sex, age groups and MCS classes, and Gray’s test was used to compare cumulative risk curves when evaluating AE events, while the log-rank test was used to compare the survival curves.

Multiple Cox regression was applied to estimate the risk of developing an AE event adjusted by sex, age groups and MCS classes. The probability of death was estimated by means of a multiple Cox regression model adjusted by sex, age groups, MCS classes and the number of AEs during the follow-up, categorized as 0, 1, 2, 3 or more and considered as time dependent. The proportional-hazard assumption was verified using the Schoenfeld residues.

Since in clinical practice IPF patients with severe disease are generally not eligible for antifibrotic treatment, patients were stratified into never-treated patients and patients receiving at least one prescription of antifibrotics in all statistical analyses.

The significance level for all the analyses was set at *p* < 0.05. Statistical analyses were performed using R, version 4.1.0.

## 3. Results

During the study period, 676 new cases of IPF were analyzed; 521 (77.1%) of the patients were from the HDR source. The median age was 75 years (IQR: 68–80); 66.6% were males and 42.3% had an MCS score between 0–4 and 57.7% equal or higher than 5. Two hundred and forty-eight patients experienced at least one AE that required hospitalization (36.7%); the two most frequent were acute respiratory failure (ICD-9-CM 518.81) and acute and chronic respiratory failure (ICD-9-CM 518.84). During the follow-up period, 271 patients (40.1%) had at least one antifibrotic prescription (Table 1).

Patients with at least one antifibrotic prescription were more frequently males, with an age <75 years, and in better health than never-treated patients (*p* < 0.001). During the follow-up, mean cumulative AE counts were significantly higher in never-treated patients than in treated patients at up to 4 years of follow-up. At the 5-year follow-up, no significant difference was observed, with an average of 80.6 subsequent AEs occurring per 100 patients in the never-treated group, compared with 75.5 subsequent AEs per 100 patients in the treated group.

### 3.1. In-Hospital Mortality

The in-hospital mortality was 8.1% (95% CI: 5.9–10.7%), i.e., 42 out of 521 patients identified through the HDR source; this proportion increased to 10.4% (95% CI: 7.6–13.8%) when calculated for never-treated patients. In-hospital deaths were more frequent in men (73.8%) with ages ≥ 75 years (88.1%) and with poor health conditions (83.3%).

### 3.2. Survival and Acute Exacerbation Free Cumulative Probability

During the follow-up we observed 276 deaths and 248 AE events, in particular 225 deaths (82%) and 175 AE events (71%), in never-treated patients.

The median time to death was 6.0 months (IQR 0.6–17.5) in never-treated patients and 21.2 months (IQR 16.4–31.6) in treated patients (*p* < 0.001); while the median time to AE was 16.3 months (IQR 7.4–27.0) and 5.4 months (IQR 2.0–13.6) in treated and never-treated patients, respectively (*p* < 0.001).

Survival and probability of not developing AE (Figure 1) were significantly higher in the group of treated patients than in never-treated patients (*p* < 0.001). After five years from the IPF diagnosis, the survival was 55.1% (95% CI: 43.3–70.1) in the former group and 34.2% (95% CI: 28.8–40.7) in the latter group; whereas the probability of being free from AE was of 51.8% (95% CI: 40.2–63.5) and of 51.5% (95% CI: 45.8–57.2), respectively in the group of treated patients and in never-treated patients.

The survival probability was significantly higher in never-treated patients of female gender, of age <75 years and in good or fair health conditions and in treated patients of age <75 years and in good or fair health conditions (Figure 2 Panel A).

No significant differences were observed in the group of never-treated and treated patients (Figure 2 Panel B).

### 3.3. Determinants of Acute Exacerbation Event

Table 2 shows the results of the multiple Cox regression model to estimate the risk of developing an AE event stratified by treatment. In never-treated patients the risk of an AE event increased by 61% in subjects with an MCS ≥5 compared to patients with an MCS between 0 and 4. No significant effects were observed in the treated group.

### 3.4. Determinants of Mortality

Table 3 shows the results of the multiple Cox regression model to estimate the risk of death stratified by treatment.

In never-treated patients, the risk of death was significantly higher in males than females, in older patients and in those patients with poor health conditions at IPF diagnosis. Compared to patients with no AE, having 1, 2 or 3 or more episodes increased the risk of death by 7.02-, 8.28- and 7.03-fold, respectively. In treated patients, older age and the rising number of AEs increased the risk of death.

### 3.5. New Cases of Pulmonary Malignant Neoplasm

Nineteen patients (12 of the never-treated patients) developed pulmonary malignant neoplasms, and 11 of these patients died during the study period. The median latency time for neoplasm was 7.1 months (IQR: 1.7–20.4), and the five-year cumulative probability of developing the cancer was 5.6% (95% CI: 2.1–9.0).

## 4. Discussion

This population-based prospective study assessed the risk of an AE event and of death in newly diagnosed IPF patients from the introduction of antifibrotics into clinical practice in Italy, providing real-world evidence.

The study showed a high mortality for all causes and risk AE in both treated and never-treated patients; in particular, the probability of death and of AE within the first year of diagnosis dramatically increased for those who have never received antifibrotic treatment. These results are consistent with those reported by Cottin et al. [23] based on an unselected cohort of untreated patients, extracted from claims data of the French National Health System between 2015 and 2016. In our study, never-treated patients were likely characterized by a more severe onset, as identified by an acute event requiring hospitalization, and were therefore not eligible for treatment. This consideration is supported by the fact that the observed in-hospital mortality in never-treated patients was 10.4%.

According to a previous study conducted in the US using secondary sources [24], patients undergoing antifibrotic therapy had a lower risk of all-cause mortality and of AE events compared with never-treated patients; treated patients maintained the survival advantage over time over never-treated patients, while the gap in the probability of AE between those treated and never-treated narrowed two years after diagnosis (Figure 1).

On the other hand, the two groups of patients had different demographic and clinical characteristics at the time of diagnosis of the disease, with never-treated patients being older and having poorer health conditions—two features that advise against the clinical decision to treat [14]. For this reason, the outcomes of interest were analyzed separately in treated and never-treated patients. The study found that male sex and poor health conditions at diagnosis negatively affects survival only in never-treated patients, while advanced age plays an important role in both treated and never-treated patients, but in the latter with greater force (Table 3). Moreover, patients with poor health conditions at diagnosis had a higher probability of AE only in the never-treated group (Table 2). AEs requiring hospitalization adversely affect survival, increasing the risk of death in both treated and never-treated patients, independently of demographic and clinical characteristics at diagnosis.

The proportion of AE events was not negligible in our study, being approximately 37% overall. This is in line with previous cohort studies that have generally reported a higher AE incidence than clinical trials [25,26,27,28,29,30].

This study has some limitations as the secondary sources do not allow the taking into account of the role of those factors related to lifestyles—such as smoking status—, to environmental or occupational exposures and to genetic background on the prognosis of the disease. Moreover, the severity of IPF cannot be assessed by administrative databases, and pulmonary function—measured by Forced Vital Capacity (FVC) or diffusing capacity for carbon monoxide (DLCO)—cannot be used as an adjustment factor. This important information could have improved the external validity of the results. Some degree of inaccuracy might be also related to timing of the first IPF diagnosis: new cases are identified through their first IPF hospitalization or their first antifibrotic prescription, both of which may, however, occur several months after an outpatient diagnosis. As a result, a proportion of incident cases identified by the HD database may have had a previous history of outpatient disease management, undetectable by the OCD. Moreover, a misclassification of AE events might be also possible, mainly due to potential overlapping with other non-respiratory causes of acute respiratory deterioration, such as cardiac failure or fluid overload. However, a wide list of ICD-9-CM codes related to upper- and lower respiratory triggers was used to identify AE events.

The study has crucial strengths, as it was well-designed, is based on an accurate multi-source case definition algorithm and a large, unselected population that shows demographic, socio-economic, and health profiles similar to those observed for the Italian population [7], analyzing a long period of 6 years.

## 5. Conclusions

This study, within the limits of an observational design based on secondary data, confirms that the combined use of health care administrative databases allows the accurate analysis of the health outcomes of a rare disease.

To our knowledge, it is the first population-based study that analyzes survival and progression of the disease using real-world data from the beginning of the antifibrotic therapy era, suggesting that timely and early diagnosis is critical to prescribing the most suitable treatment to increase survival and maintain a healthy life expectancy.

## Figures and Tables

**Figure 1 ijerph-19-16689-f001:**
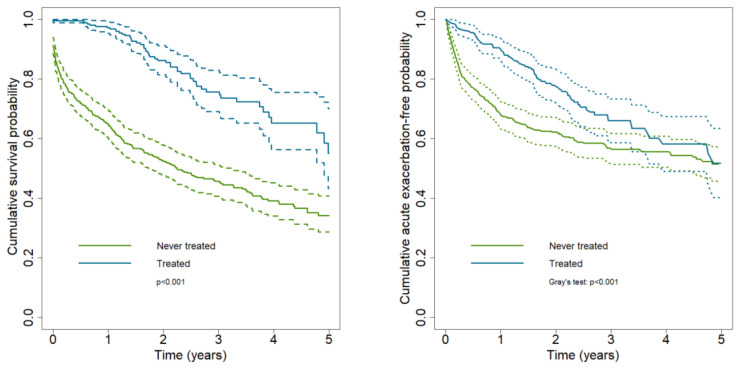
Kaplan–Meier cumulative survival and acute exacerbation-free probability curves according to treatment status (Treated: patients with at least one prescription of antifibrotics during the follow-up were considered as treated). The cumulative acute exacerbation-free probability is calculated considering death as a competing risk event. (Dashed line: 95% confidence band).

**Figure 2 ijerph-19-16689-f002:**
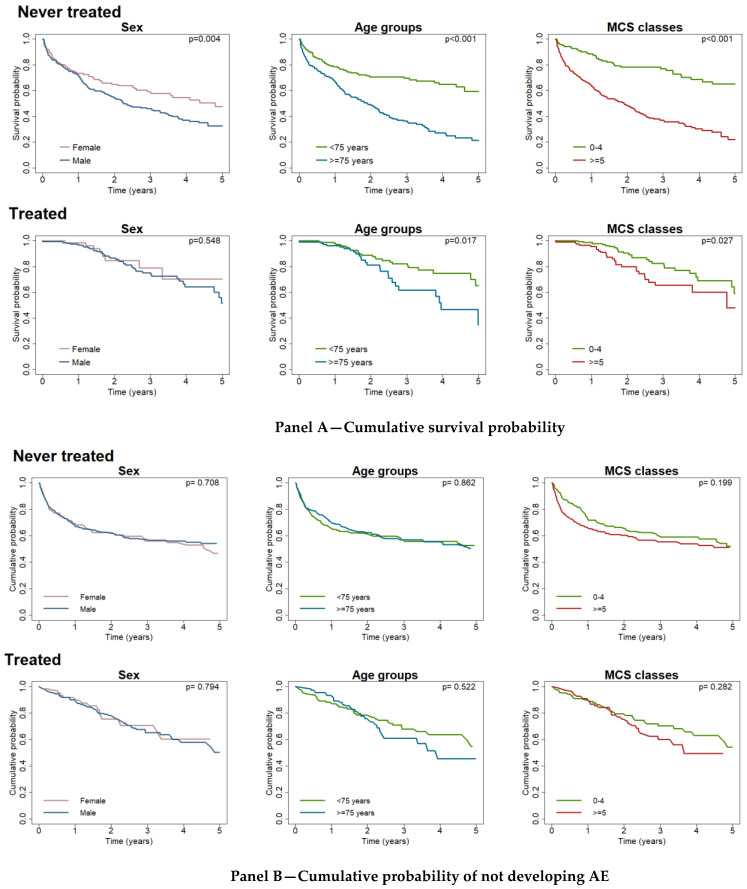
Cumulative survival probability (**Panel A**) and probability of not developing Acute Exacerbations (**Panel B**) by sex, age groups and Multisource Comorbidity Score (MCS) classes in not treated and treated patients (Treated: patients with at least one prescription of antifibrotics during the follow-up were considered as treated). The cumulative acute exacerbation-free probability is calculated considering death as competing risk event.

**Table 1 ijerph-19-16689-t001:** Subjects’ characteristics at IPF diagnosis according to treatment.

	Total	Never-Treated	Treated	*p*
	n (%)	n (%)	n (%)	
**Incident cohort**	676 (100%)	405 (100%)	271 (100%)	
**Sex**				<0.001 ^#^
Female	226 (33.4%)	163 (40.2%)	63 (23.2%)	
Male	450 (66.6%)	242 (59.8%)	208 (76.8%)	
**Median age (IQR) in years**	75 (68–80)	77 (67–83)	74 (69–77)	<0.001 *
**Age group**				<0.001 ^#^
<75 years	325 (48.1%)	166 (41%)	159 (58.7%)	
≥75 years	351 (51.9%)	239 (59%)	112 (41.3%)	
**MCS classes**				<0.001 ^#^
MCS 0–4	286 (42.3%)	129 (31.9%)	157 (57.9%)	
MCS ≥5	390 (57.7%)	276 (68.1%)	114 (42.1%)	
**Mean Cumulative Counts of AEs per 100 patients at**				
1 year	48.3	56.7	33.1	0.028 °
2 years	60.1	67.0	49.3	0.013 °
3 years	68.7	73.3	61.3	0.027 °
4 years	72.5	75.6	68.9	0.048 °
5 years	78.7	80.6	75.5	0.080 °

**Treated**: Patients with at least one prescription of antifibrotics during the follow-up were considered as treated. **IQR**: interquartile range; **MCS**: Multisource Comorbidity Score; **AEs**: Acute exacerbations. ^#^ Chi-square test; * Mann–Whitney test; ° Difference between the AUCs of mean cumulative count curves.

**Table 2 ijerph-19-16689-t002:** Cox proportional hazard model for the risk of acute exacerbation. The analyses were stratified according to treatment.

	Never-Treated			Treated		
	HR	95% CI	*p*	HR	95% CI	*p*
**Sex (Female r.c.)**						
Male	0.98	0.73–1.34	0.922	1.12	0.61–2.07	0.711
**Age groups (<75 r.c.)**						
≥75 years	1.25	0.91–1.71	0.168	1.14	0.70–1.87	0.593
**MCS classes (0–4 r.c.)**						
≥5	1.61	1.14–2.26	0.006	1.32	0.82–2.12	0.261

**Treated**: patients with at least one prescription of antifibrotics during the follow-up were considered as treated; **MCS**: Multisource Comorbidity Score; **r.c.**: reference category; **HR**: Hazard Ratio; **95% CI**: 95% confidence interval.

**Table 3 ijerph-19-16689-t003:** Cox proportional hazard model for the risk of death. The number of AEs has been considered as a time-dependent variable. The analyses were stratified according to treatment.

	Never-Treated			Treated		
	HR	95% CI	*p*	HR	95% CI	*p*
**Sex (Female r.c.)**						
Male	1.43	1.08–1.90	0.012	1.68	0.76–3.71	0.202
**Age groups (<75 r.c.)**						
≥75 years	2.14	1.58–2.91	<0.001	1.88	1.07–3.31	0.029
**MCS classes (0–4 r.c.)**						
≥5	2.24	1.58–3.17	<0.001	1.46	0.83–2.57	0.190
**N. of AE (no AE r.c.)**						
1	7.02	4.92–10.03	<0.001	16.95	7.80–36.84	<0.001
2	8.28	5.19–13.20	<0.001	16.54	6.93–39.48	<0.001
3 or more	7.03	3.93–12.58	<0.001	17.90	7.34–43.68	<0.001

**Treated**: patients with at least one prescription of antifibrotics during the follow-up were considered as treated; **MCS**: Multisource Comorbidity Score; **AE**: Acute Exacerbation; **r.c.**: reference category; **HR**: Hazard Ratio; **95% CI**: 95% confidence interval.

## Data Availability

Restrictions apply to the availability of these data. Data was obtained from the Marche Region and are available with the permission of Marche Region.

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
