# Peer review of "Prognosis and Survival in Idiopathic Pulmonary Fibrosis in the Era of Antifibrotic Therapy in Italy: Evidence from a Longitudinal Population Study Based on Healthcare Utilization Databases"

_ijerph, 2022, doi:10.3390/ijerph192416689_

Round 1

Reviewer 1 Report

I have just two observations on the statistical analysis part:

1 - In Table 1 the number of AE events are reported according to treatment; I think that since these numbers could vary according to the follow up times, it would be better to report AE incidence rates at different time points (or mean cumulative counts);

2-  Related to the previous observation: if death could be a competing risk with respect to AE event, I think that it is better to estimate the cumulative incidences curves of AE events taking into account death as a competing event (Figure 1 and Figure 2 panel b)

Author Response

To Editor: Thank you for giving us the opportunity to submit a revised draft of the manuscript “Prognosis and survival in idiopathic pulmonary fibrosis in the era of antifibrotic therapy in Italy: latest evidence from the real world” for publication in the International Journal of Environmental Research and Public Health.

We thank the reviewers for their time in reviewing our manuscript and for their comments and suggestions useful to improve our article.

We would like to underline that the results presented in this manuscript were not obtained from a clinical study but from a study based on the use of secondary data sources. Therefore, we were not able to follow some of the suggestions of Reviewer 2, because she/he requested clinical information (such as results of laboratory tests or diagnostic/surgical procedure, smoking habits) not recorded/collected/available in healthcare administrative databases.

We provide the point-by-point responses. All modifications in the manuscript and supplementary materials have been marked up using the “Track Changes” MS Word function.

Response to Reviewer 1 Comments

Point 1: In Table 1 the number of AE events are reported according to treatment; I think that since these numbers could vary according to the follow up times, it would be better to report AE incidence rates at different time points (or mean cumulative counts);

Response 1: Thanks for your suggestion. We estimated the mean cumulative count curves, reporting values at 1 up to 5 years of follow-up by treatment in table 1.We have modified the statistical analysis section as follows, including 2 new references: “Mean cumulative counts at 1 up to 5 years of follow-up were estimated to evaluate the total burden of subsequent AEs among IPF patients by treatment [21], considering death as a competing risk event; comparisons of mean cumulative count curves were performed via area under the curve (AUC) [22].”

We have also modified the result section as follows: “During the follow-up, mean cumulative AE counts were significantly higher in patients never treated than in treated patients up to 4 years of follow-up. At 5-year follow-up, no significant difference was observed, with an average of 80.6 subsequent AEs occurring per 100 patients in the never treated group, compared with 75.5 subsequent AEs per 100 patients in the treated group.”

Point 2: Related to the previous observation: if death could be a competing risk with respect to AE event, I think that it is better to estimate the cumulative incidences curves of AE events taking into account death as a competing event (Figure 1 and Figure 2 panel b).

Response 2: Thanks for this comment. We have performed the competing risk analysis to evaluate the probability of AE events and modified Figure 1 and Figure 2 panel b, and the probabilities reported in the results section. We modified the section of statistical analysis as follows (Page 3): “When evaluating AE event, death was considered as competing risk event” […] “All patients were followed up for at least 180 days. Kaplan-Meier curves were estimated stratifying by sex, age groups, and MCS classes and Gray's test was used to compare cumulative risk curves when evaluating AE event, while the Log-rank test was used to compare the survival curves.”

We have also modified the results section (Page 6) as follow: “The probability of not developing AE was significantly higher in never treated patients younger than 75 years old or with a MCS score between 0 and 4. No significant differences were observed in the group of never treated and treated patients (Figure 2 Panel B).”

Reviewer 2 Report

The authors present an original study concerning survival of IPF patients in the era of the antifibrotic agents. They compare a group of patients that receives antifibrotic treatment with an untreated one. However, I am concerned about major issues.

Major issues:

1.     The study is based on administrative data, therefore selection bias may have been made. How IPF was defined? Did all patients present UIP pattern in HRCT? Did patients belong to definite or probable UIP group?

2.     In your cohort, what percentage had performed surgical biopsy? Did you control if the surgical biopsy affected the survival?

3.     With what criteria the patients received or not antifibrotic treatment? Selection and information bias are possible.

4.     In the methods, you state that Acute exacerbations of IPF in your study were defined as any acute respiratory-related hospitalization but these may not be ‘real’ episodes of acute exacerbations. Acute exacerbation is a devastating event with high mortality. It is not often for patients to have 3 and 4 episodes of AE-IPF. Maybe some events are misclassified and were due to viral infections, or respiratory insufficiency or cardiac failure.

5.     How did the authors conclude that these episodes were real acute exacerbation? Did they measure pro-BNP or did patients had performed heart ultrasound?

6.     Did the patients perform CT pulmonary angiogram to exclude pulmonary embolism in order to fulfil the definition of AE-IPF?

7.     Smoking status is a very important information in the clinical characteristics of your patients. I should advise you to include it in Table 1.

8.     The authors have not included lung capacity measures of treated and untreated patients, including FVC and DLCO in order to see the differences between the groups. Lung capacity has a critical prognostic role and may have also affected the survival of each group.

9.     Could less severe cases not have been included in the administrative data?

10.  You state in the results that male sex negatively affected survival but this was not a significant result. Please comment.

Minor Issues:

Line 1 in the introduction, please correct ‘devastating’.

Author Response

To Editor: Thank you for giving us the opportunity to submit a revised draft of the manuscript “Prognosis and survival in idiopathic pulmonary fibrosis in the era of antifibrotic therapy in Italy: latest evidence from the real world” for publication in the International Journal of Environmental Research and Public Health.

We thank the reviewers for their time in reviewing our manuscript and for their comments and suggestions useful to improve our article.

We would like to underline that the results presented in this manuscript were not obtained from a clinical study but from a study based on the use of secondary data sources. Therefore, we were not able to follow some of the suggestions of Reviewer 2, because she/he requested clinical information (such as results of laboratory tests or diagnostic/surgical procedure, smoking habits) not recorded/collected/available in healthcare administrative databases.

We provide the point-by-point responses. All modifications in the manuscript and supplementary materials have been marked up using the “Track Changes” MS Word function.

Response to Reviewer 2 Comments 

Point 1: The study is based on administrative data, therefore selection bias may have been made. How IPF was defined? Did all patients present UIP pattern in HRCT? Did patients belong to definite or probable UIP group?

Response 1: To answer the questions “How IPF was defined?”, as reported in the paragraph “Study population, data sources, incident IPF case definition”, we use one of the most frequently definition used in scientific literature on IPF epidemiology, when using healthcare administrative databases. In a previous study on IPF incidence [Iommi et al, 2022 – reference number 7 in the manuscript] we compared three different algorithms suggested by G. Raghu et al (“Incidence and Prevalence of Idiopathic Pulmonary Fibrosis,” Am. J. Respir. Crit. Care Med., vol. 174, no. 7, pp. 810–816, Oct. 2006, doi: 10.1164/rccm.200602-163OC) and S. Harari et al (“Epidemiology of Idiopathic Pulmonary Fibrosis in Northern Italy,” PLoS One, vol. 11, no. 2, p. e0147072, Feb. 2016, doi: 10.1371/journal.pone.0147072) and as a result the General Case Definition was the one with the highest accuracy in identifying new Cases of IPF (Iommi et al, 2022 – reference number 7 in the manuscript), explored through a validation with clinical data retrieved from one of the hospital included.

To answer the questions “Did all patients present UIP pattern in HRCT? Did patients belong to definite or probable UIP group?”, as previously stated, IPF cases were derived from administrative healthcare databases using a predefined algorithm involving the specific ICD-9-CM Code referred to IPF diagnosis (i.e. 516.3) and antifibrotic prescriptions. This was not a “classical” clinical, hospital-based observational study, thus, clinical information on HRCT pattern, serological findings, and functional data, were not available, because not reported in the databases we used.

Point 2: In your cohort, what percentage had performed surgical biopsy? Did you control if the surgical biopsy affected the survival?

Response 2: Surgical lung biopsy (ICD-9-CM codes 33.28) or transbronchial lung biopsy (ICD-9-CM code 33.27) in hospitalization or outpatient visits, were performed after diagnosis by overall 23 patients (3.4%), 13 never treated (3.2%), 10 treated (3.7%), and no significant effect on survival was observed.

Point 3: With what criteria the patients received or not antifibrotic treatment? Selection and information bias are possible.

Response 3: It has been assumed that the eligibility criteria reported by Italian regulatory agency (Agenzia Italiana del Farmaco-AIFA) were followed for the prescription of antifibrotic drugs, as stated by the co-author clinicians of this manuscript (MB, FM), who work in one of the major centres of reference for the diagnosis and care of IPF in the Marche Region. In Italy, pulmonologists allowed to prescribe antifibrotics, have to fill in, at time of first prescription, a specific request form to AIFA, reporting the eligibility criteria, in order to obtain the authorization. In fact, in the statistical analysis paragraph, page 4, lines 146-148, we stated that: “Since in clinical practice IPF patients with a severe disease are generally not eligible for antifibrotic treatment, patients were stratified in never treated and in patients receiving at least one prescription of antifibrotics in all statistical analyses”.

Moreover, in the discussion section, page 9, lines 244-246 we stated that: “In our study, never treated patients were likely characterized by a more severe onset, as identified by an acute event requiring hospitalization, therefore not eligible for treatment.”

Point 4: In the methods, you state that Acute exacerbations of IPF in your study were defined as any acute respiratory-related hospitalization but these may not be ‘real’ episodes of acute exacerbations. Acute exacerbation is a devastating event with high mortality. It is not often for patients to have 3 and 4 episodes of AE-IPF. Maybe some events are misclassified and were due to viral infections, or respiratory insufficiency or cardiac failure.

Response 4: We were aware that some AE events might be misclassified, due to a potential overlapping with cardiac failure, or fluid overload, and this limitation was stated in the discussion as follows: “a misclassification of AE events might be also possible, mainly due to the potential overlapping with other non-respiratory causes of acute respiratory deterioration, such as cardiac failure or fluid overload. However, a wide list of ICD-9-CM codes related to upper- and lower respiratory triggers was used to identify AE events.” 

Anyway, patients with a diagnosis of cardiac failure in primary field of the HDR database were not counted as AE. Regarding viral infections, by study protocol, we did not exclude patients reporting codes referring to these in the primary/secondary fields of the HDR database, thus, some AE “triggered” by viral infections can be present in our cohort. However, according to the latest expert statement on AE definition, it is no longer needed to exclude viral trigger to define an event as AE. The proposed definition is: “any acute respiratory event characterized by new bilateral ground-glass opacification/consolidation not fully explained by cardiac failure or fluid overload” (Collard et at, AJCCM 2016).

Point 5: How did the authors conclude that these episodes were real acute exacerbation? Did they measure pro-BNP or did patients had performed heart ultrasound?

Response 5: We defined acute exacerbation as any acute respiratory-related hospitalization occurred after the index date, since the results of laboratory tests or diagnostic procedure are not available in secondary sources in Marche Region. We described AE definition in the Outcomes paragraph, page 3, lines 112-113, showing in Table S2 the list of ICD-9-CM codes that were searched in the primary or secondary diagnoses fields of the HDR database.

Point 6: Did the patients perform CT pulmonary angiogram to exclude pulmonary embolism in order to fulfil the definition of AE-IPF?

Response 6: As previously stated, specific clinical information on imaging were not available, because not reported in the databases we used. The codes related to pulmonary embolism were not included in the Table S2 list.

Point 7: Smoking status is a very important information in the clinical characteristics of your patients. I should advise you to include it in Table 1.

Point 8: The authors have not included lung capacity measures of treated and untreated patients, including FVC and DLCO in order to see the differences between the groups. Lung capacity has a critical prognostic role and may have also affected the survival of each group.

Responses 7 and 8: When healthcare administrative databases are used for epidemiologic purpose, such as hospital discharge records and drug prescription databases, a lot of clinical information are not available, because they are not collected, such as FVC and DLCO and smoking habits. These study limitations were faced in the discussion section (page 10, lines 267-276) as well as the strengths of studies based on secondary data sources (page 2, lines 67-69; page 10, lines 281-284).

Point 9: Could less severe cases not have been included in the administrative data?

Responses 9: We agree with this comment, in fact, in the discussion section among the limitation of the study (page 10, lines 274-276), we stated that: “As a result, a proportion of incident cases identified by the HD database may have had a previous history of outpatient disease management, undetectable by OCD.”

However, less severe cases are less likely to be hospitalized but more likely to receive antifibrotics, so we detected them by drug prescriptions, while, on the other hand, more severe cases are less likely to receive the antifibrotic drugs (due to the lack of eligibility criteria) but more likely to be hospitalized. Therefore, the expected proportion of cases not identified is deemed as not significant.

Point 10: You state in the results that male sex negatively affected survival but this was not a significant result. Please comment.

Responses 10: Thank you for this comment. We have mistakenly copied other values into the table. It is now updated with the correct results.

Point 11: Line 1 in the introduction, please correct ‘devastating’.

Responses 11: Thank you. We have corrected the term.

Round 2

Reviewer 2 Report

I would like to thank the authors for addressing some of our comments. However, I think that selective and information bias limit the significance of the results of the study. Lack of information data regarding smoking and DLCO are important factors for the disease outcome that may have affected the results. Also, as stated by the authors, the fact that most severe cases did not receive antifibrotics affects the distribution of the 'severe cases' between the two groups, therefore the results maybe confounded. The lack of DLCO as an information does not permit us also to adjust for this value that would be an indicator of the severity of the pulmonary fibrosis. For these reasons i would propose to reject the publication.

Author Response

We thank the Reviewer for his/her precious time in reviewing our paper. We have better explained the limitations of the study, emphasizing that the lack of important information (such as smoking status, FVC or DLCO) limits the external validity of the results.